# Understanding the Transmission Dynamics of the Chikungunya Virus in Africa

**DOI:** 10.3390/pathogens13070605

**Published:** 2024-07-22

**Authors:** Yajna Ramphal, Houriiyah Tegally, James Emmanuel San, Martina Larissa Reichmuth, Marije Hofstra, Eduan Wilkinson, Cheryl Baxter, Tulio de Oliveira, Monika Moir

**Affiliations:** 1Centre for Epidemic Response Innovation (CERI), School for Data Science and Computational Thinking, Stellenbosch University, Stellenbosch 7600, South Africa; 26700689@sun.ac.za (Y.R.); houriiyah@sun.ac.za (H.T.); marijeh@sun.ac.za (M.H.); ewilkinson@sun.ac.za (E.W.); cbaxter@sun.ac.za (C.B.); 2Duke Human Vaccine Institute, Duke University, Durham, NC 27710, USA; sanemmanueljames@gmail.com; 3Institute of Social and Preventive Medicine (ISPM), University in Bern, 3012 Bern, Switzerland; martina.reichmuth@unibe.ch; 4KwaZulu-Natal Research Innovation and Sequencing Platform (KRISP), University of KwaZulu-Natal, Durban 4001, South Africa

**Keywords:** Chikungunya virus, Africa, transmission dynamics, epidemiology, genomic distribution, genomic surveillance

## Abstract

The Chikungunya virus (CHIKV) poses a significant global public health concern, especially in Africa. Since its first isolation in Tanzania in 1953, CHIKV has caused recurrent outbreaks, challenging healthcare systems in low-resource settings. Recent outbreaks in Africa highlight the dynamic nature of CHIKV transmission and the challenges of underreporting and underdiagnosis. Here, we review the literature and analyse publicly available cases, outbreaks, and genomic data, providing insights into the epidemiology, genetic diversity, and transmission dynamics of CHIKV in Africa. Our analyses reveal the circulation of geographically distinct CHIKV genotypes, with certain regions experiencing a disproportionate burden of disease. Phylogenetic analysis of sporadic outbreaks in West Africa suggests repeated emergence of the virus through enzootic spillover, which is markedly different from inferred transmission dynamics in East Africa, where the virus is often introduced from Asian outbreaks, including the recent reintroduction of the Indian Ocean lineage from the Indian subcontinent to East Africa. Furthermore, there is limited evidence of viral movement between these two regions. Understanding the history and transmission dynamics of outbreaks is crucial for effective public health planning. Despite advances in surveillance and research, diagnostic and surveillance challenges persist. This review and secondary analysis highlight the importance of ongoing surveillance, research, and collaboration to mitigate the burden of CHIKV in Africa and improve public health outcomes.

## 1. Introduction

The Chikungunya virus (CHIKV) is an arthritogenic, enveloped, positive-strand RNA Alphavirus, belonging to the family Togaviridae, that is responsible for the febrile Chikungunya illness. Clinical studies from the eighteenth and nineteenth centuries, along with molecular clock analyses of contemporary CHIKV genomes, suggest that this virus existed for 300 to 500 years before the first isolation in 1953 [1,2]. The virus is globally widespread in tropical and subtropical regions and is transmitted by *Aedes* mosquito species, with *Ae. albopictus* and *Ae. aegypti* being the primary vectors for transmitting CHIKV to humans [3,4,5,6]. CHIKV is classified into four lineages: East–Central–South Africa (ECSA), Asian, Indian Ocean lineage (IOL), and West African (WA). Recent investigations have shown that the distribution of *Ae. albopictus* has expanded to locations with lower temperatures, raising concerns about the potential for CHIKV transmission in new climatic environments [5,6].

The Chikungunya fever outbreak in the Newala district of Tanzania in 1952 and 1953 holds notable significance, as it suggests some of the earliest documented cases in humans and marks the first isolation of the virus [4,7,8]. The term “Chikungunya” is derived from the Kimakonde language, signifying “to bend up” or “to become contorted”, which encapsulates the severe joint pain and altered posture, hallmark symptoms of the disease [6,8]. The outbreak was characterized by a high prevalence of cases, accompanied by reports of febrile illness and incapacitating joint pain of varying severity [6,7]. Nineteenth-century records document an epidemic termed Kidenga pepo in Zanzibar during 1823, a Swahili term denoting “a sudden cramp-like seizure caused by an evil spirit”, spreading to the Americas and West Indies in 1827 [9]. Prior to this, evidence points to an earlier epidemic spanning from 1779 to 1785, affecting Egypt, the Arabian Peninsula, India, and Southeast Asia [9,10].

CHIKV infections vary widely in severity, from mild to severe symptoms, with 3–28% of cases being asymptomatic [11]. The principal indicators include acute fever and profound joint pain [12]. Concomitant symptoms during the acute phase of the disease encompass myalgia, headache, fatigue, and gastrointestinal disturbances. Within a few days following the onset of fever, individuals may develop a maculopapular rash spanning the trunk, limbs, and face [5,13]. While the majority of cases recover within weeks, some individuals, particularly those with underlying health conditions, may exhibit self-limiting prolonged joint pain and fatigue. Severe complications, although rare, can include neurological and cardiovascular manifestations [5]. High rates of mother-to-child transmission have been observed during outbreaks in India, the Americas, and Réunion Island, leading to neonatal disease with significant impacts on infant health. Recent evidence from a cohort study in Nigeria (2019–2022) has highlighted significant associations between CHIKV infections during pregnancy and adverse birth outcomes [14,15]. The arthritic symptoms of Chikungunya fever can persist for several years [3]. Factors like age, duration of infection, joint pain, and swelling adversely impact the health and quality of life, leading to pervasive absenteeism and economic losses. For example, in the Réunion Island outbreak of 2005–2006, the total medical expenditure of the chikungunya epidemic was estimated to be EUR 43.9 million, with direct healthcare expenses and the loss of productivity estimated at EUR 26.5 million and EUR 17.4 million, respectively [16,17].

Clinical presentation of CHIKV infection may vary depending on the geographic location of the epidemic. This variability may be attributed to factors such as genetics, comorbidities, pre-existing immunity to CHIKV, and socioeconomic status of the affected communities [13]. For instance, Paixåo et al. (2018) [18] investigated the association of CHIKV genotype heterogeneity with disease severity. The study found considerable variability in the prevalence of self-reported chronic symptoms with CHIKV lineages. The prevalence of chronic disease was significantly higher in IOL (52%) followed by the Asian (39%) and ECSA (14%) lineage.

In low-resource socioeconomic settings, the prevalence of undifferentiated febrile illness (UFI) outbreaks poses a significant diagnostic challenge [19]. Over the last decade, many African nations, including Gabon, Nigeria, Kenya, the Democratic Republic of Congo, and Sudan, have witnessed substantial outbreaks of febrile diseases [20]. These epidemics are frequently associated with long-term disabilities and high case fatality rates [21,22]. A systematic review and meta-analysis by Nooh et al. (2023) [22] revealed that the aetiology of febrile illness is unknown in more than 60% of febrile patients in East Africa, with CHIKV identified as the most prevalent causative pathogen in the analysed studies.

Understanding the epidemiology, clinical impact, and socioeconomic implications of CHIKV in Africa is vital for effective disease management and control. The Climate Amplified Diseases and Epidemics (CLIMADE) consortium, from which this work stems, was established to develop approaches to predict, track, and control diseases such as CHIKV in the regions most impacted by climate change (https://climade.health/africa, accessed on 5 June 2024). This initiative, with global collaborators, aims to form a pan-African coalition to create localised knowledge and technologies to fill crucial knowledge gaps in disease transmission, improve the genomic representation of climate-amplified diseases, and develop sustained genomic sequencing capacities [23].

The transmission dynamics and burden of CHIKV in low-resourced regions in Africa remain poorly understood. Our study combines a literature review with phylogenetic and phylodynamic analyses of publicly available genomic data to provide insights into the spatiotemporal movement within and out of the African continent. We review and synthesize the current knowledge on CHIKV in Africa, aiming to elucidate the genetic diversity and evolutionary dynamics while identifying factors contributing to the spread of infection, and highlighting existing burdens and knowledge gaps. We hypothesize that heterogeneous transmission dynamics across different regions in Africa have driven the widespread circulation of CHIKV.

## 2. Cases, Outbreaks, and Current Burden in Africa

Since its discovery in 1953, Africa has witnessed numerous CHIKV outbreaks, with an estimated 70% of the population residing in high-risk areas for arboviral infections [24]. Suspected CHIKV cases are defined by the presence of acute high fever (>38.5 °C) and severe joint pain. Laboratory confirmation requires CHIKV isolation, RNA detection by RT-PCR, CHIKV-specific IgM antibodies, or an increase in IgG antibodies over time [25]. Outbreaks can be substantial in areas with high mosquito vector activity and favourable environmental conditions for viral transmission [26].

Figure 1 illustrates the distribution and surveillance of CHIKV in Africa based on the following data sources: World Health Organization African Regional Office (WHO-AFRO) CHIKV reported cases since 2018 (https://www.afro.who.int/health-topics/disease-outbreaks/outbreaks-and-other-emergencies-updates, accessed on 28 April 2024), publicly available genomic data (https://www.ncbi.nlm.nih.gov/nuccore, accessed on 15 January 2024), case and outbreak data from the published literature. Figure 1 highlights the disparities between reported cases and genomic data. Ethiopia and Chad reported over 50,000 and 30,000 suspected cases in 2019 and 2020, respectively, without corresponding genomic data, suggesting surveillance gaps. Additionally, the distinct differences in genotype circulation in West, Central, and East Africa become immediately apparent.

In Central Africa, CHIKV was first detected in 1958 in the Doruma region in the DRC [63]. Significant outbreaks occurred in 1999 and 2000 in Brazzaville and Kinshasa, resulting in an estimated 50,000 reported cases [60]. Outbreaks in 2011 and 2018 saw cases reaching 8000 and 11,000, respectively. From 2002 to 2006, isolated CHIKV outbreaks were recorded in Equatorial Guinea, Sudan, and Cameroon. Cameroon reported cases throughout this period [12,30,64]. In 2007, Gabon experienced a CHIKV outbreak that impacted at least 20,000 individuals in Libreville, the country’s capital and largest city. This outbreak lasted 3 years and extended to southern areas of the country [12].

The West African region emerges as a potential hotspot for arboviral disease transmission, with significant outbreaks in Senegal, Nigeria, and intermittent cases in Guinea in 1992 and 2002, Sierra Leone in 2012–2013, and Mali and Burkina Faso in 2023–2024 (Figure 1) [6,50,65,66,67,68,69,70,71]. A recent study on arboviruses in Southern Mali revealed a CHIKV seropositivity rate of 31.2% post-rainy season in 2015 [70]. Two genomes were isolated from Cote d’Ivoire in 1981 and 1993 [71].

In East Africa, Kenya witnessed an unprecedented epidemic in 2004 spreading to Mombasa and Lamu Island. Lamu Island reported a 75% attack rate (AR) [72,73]. By 2005, the virus spread to the islands of Comoros, Mayotte, Seychelles, Réunion, and Mauritius. Réunion Island alone reported an estimated 250,000 cases, affecting about 32% of the island’s population [35]. After a 12-year hiatus in Kenya, unprecedented CHIKV outbreaks occurred in Mandera City in 2016 and Mombasa in 2017–2018 [6,35]. In neighbouring East African countries, Tanzania reported CHIKV infections between 2007 and 2008. Somalia reported its first cases in 2016 and Ethiopia reported more than 50,000 suspected cases in 2019 [6]. During the nine-month epidemic in Sudan from 2017 to 2018, Kassala State and Red Sea State collectively reported 48,763 cases [36].

Efforts were made after the 2005 outbreak on Réunion Island to differentiate between the two epidemic waves. The second wave exhibited a notably higher AR compared to the initial wave and a higher proportion of symptomatic cases relative to total infections. The severity of symptoms and incidence of neuro-chikungunya was also greater during the second wave, with a higher proportion of children discharged with disabilities. These findings underscore the evolving nature and severity of the epidemic across distinct phases, highlighting the need for comprehensive surveillance and response strategies within regions of high transmission risk and their neighbouring countries [73,74].

## 3. Genotype Distribution

Since the 1950s, over 110 countries have reported CHIKV outbreaks. The virus, with origins in Sub-Saharan Africa, has spread to new territories, including Europe and the Pacific, largely facilitated by global travel and trade [11,37,51,75,76] (Figure 2). This widespread dissemination has led to the emergence of three monophyletic and geographically distinct genotypes [5].

CHIKV is a single-stranded, positive-sense RNA virus, which has evolved as a quasispecies with geographically distinct variants [75]. Since the first identification in East Africa in 1953, the ECSA genotype initially circulated in east, southern, and central regions of Africa and has since spread to cause significant outbreaks in Asia and the Americas [77]. The ECSA genotype comprises three clades: Clade 1 (ancestral East and Southern Africa strains), Clade 2 (Central African strains), and Clade 3 (Indian Ocean lineage), with origins in Eastern Africa, disseminating to Indian Ocean islands, subsequently reaching Southeast Asia, Oceania, and Europe (Figure 2a) [6].

The re-emergence of CHIKV in 2004 triggered outbreaks in the Indian Ocean basin, leading to the emergence of IOL, characterized by higher vector competence in *Ae. albopictus* [78]. Genomic data suggest that IOL initially spread to the Comoros and Réunion islands, rapidly expanding and diversifying, causing outbreaks in the Indian Ocean islands and later spreading to India, Sri Lanka, Thailand, Malaysia, and Italy [35,77,78].

In 2013, a significant milestone in CHIKV transmission occurred when the Asian lineage was imported into St. Martin’s Island, triggering ongoing outbreaks across the Americas [79]. The virus subsequently spread to neighbouring countries driven by international travel, trade, and the presence of suitable mosquito vectors [80,81]. Additionally, the ECSA lineage was introduced in the Bahia state of Brazil in 2014, resulting in over 5000 reported cases. Since then, over 3.6 million cases have been reported in the Americas [79,82,83]. The cocirculation of the ECSA and Asian genotypes have led to outbreaks in South America and the Caribbean islands since 2014, highlighting the global spread and adaptability of CHIKV genotypes [83].

The WA lineage of CHIKV was first identified in 1964 in Ibadan, Nigeria, following its isolation from a mosquito sample obtained in 1963 [68]. Subsequently, it was discovered in Rufisque, Senegal, in 1966 [65]. The ECSA lineage has also been detected in West Africa, isolated from a bat sample in Senegal in 1963, suggesting dissemination beyond its geographical origin and possible spatial overlap of the two lineages. The potential coexistence of the WA and ECSA genotypes in West Africa emphasizes the region’s complex CHIKV dynamics; however, this was a singular occurrence with no ECSA lineage strains detected since [75].

## 4. Importance of Genomic and Epidemiological Surveillance

Genomic surveillance is crucial for understanding the transmission dynamics, evolution, and epidemiology of vector-borne diseases. Traditionally, public health officials relied extensively on epidemiological data, but advancements in whole-genome sequencing and phylogenetic methods have enhanced the ability to map transmission pathways, detect viral strain, and characterize lineages [84]. Integrating clinical, demographic, and epidemiological metadata with sequencing data helps to identify variables that influence transmission patterns, viral competence, lineage prevalence, and epidemic potential of emergence strains [3,80,85,86]. For example, during the SARS-CoV-2 pandemic, global genomic and epidemiological surveillance revolutionized variant tracking and transmission across spatial and temporal scales and elucidated the multifactorial nature of virus evolution [86].

Urbanization, human mobility, and climate change have facilitated the movement of arboviruses to new regions, increasing the risk of outbreaks within previously unexposed populations, in turn highlighting the need for effective surveillance systems [85]. Comprehensive genetic data from each outbreak and interepidemic strains is essential for tracking the movement of viral strains and identifying mutations that may contribute to future outbreaks [72,86].

Since 2016, a flagship arbovirus genomic and epidemiological surveillance program in South America has characterized CHIKV outbreaks and epidemics in Brazil [87], Uruguay [88], Paraguay [89], and Argentina [79], demonstrating the crucial role of genomic surveillance for vector-borne diseases. In contrast, Africa has lacked a comparable program, and genomic data from African CHIKV cases remain limited despite the virus being endemic in many regions. CLIMADE’s strategy for Africa includes forming a consortium to develop locally relevant knowledge and technologies, thereby addressing major knowledge gaps in disease transmission. Key goals include forming a pan-African coalition to assess arbovirus susceptibility, improving genomic representation of climate-amplified diseases, and developing long-term genomic capability and collaboration with global partners [23].

Currently, there are 393 CHIKV genomes sampled from Africa available publicly, comprising 93 whole genomes and 300 partial gene sequences (Figure 3). Advances in sequencing technologies have shifted from Sanger sequencing to next-generation sequencing platforms such as Illumina and Oxford Nanopore for whole-genome sequencing (Appendix A).

Figure 3 highlights the evolving landscape of CHIKV sequencing in Africa, showing the steady increase in genomic data generation. CHIKV sequencing efforts began in the early 1950s, with a notable uptick observed since the early 2000s. Predominant sources of genomic data include *Aedes* mosquito vectors and human hosts, while genomes have also been isolated from bats and mice. The ECSA genotype appears consistently over decades, with the WA genotype appearing less frequently and sporadically since the 1960s. A notable increase in IOL genomes between 2010–2020 reflects the increase in resources directed to sequencing efforts after major outbreaks. Continued investment in routine genomic surveillance is crucial for the characterization and tracking of epidemic strains. This effort provides essential information for implementing effective control and preventive measures against CHIKV and other vector-borne diseases in Africa.

## 5. Genetic Diversity and Transmission Dynamics in Africa

The transmission dynamics of CHIKV in Africa are shaped by entomological, environmental, and sociodemographic factors [2]. Our analysis (see Appendix A for a description of the methods) reveals geographically distinct lineages, including the WA, the ECSA, and the IOL (Figure 4), exhibiting distinct genetic fingerprints and transmission patterns [80]. Estimated evolutionary rates for each lineage varied considerably. The highest rate was observed in the IOL of 5.31 × 10^−4^ substitutions per nucleotide per year (subs/nt/y) (Figure 4a–c), followed by the ECSA, Asian, and WA lineage of 4.44 × 10^−4^, 4.178 × 10^−4^, and 2.84 × 10^−4^ subs/nt/y, respectively. A study conducted by Volks et al. in 2010 obtained comparable evolutionary rates, which showed a notable difference between enzootic lineages having a lower evolutionary rate than epidemic lineages [75]. Here, we included South American ECSA strains in the phylogenetic analysis, which likely contributed to the difference in rates observed in the aforementioned study.

Geographic and environmental factors such as seasonality, vector abundance, and human mobility significantly influence CHIKV evolution [2,5,90]. For instance, ECSA sublineages 1 and 2 observed in East, Southern, and Central African clades (Figure 4a) show long-standing regional diversity, suggestive of limited genomic exchange and localized transmission [6,75]. Similarly, the monophyletic clustering of WA sequences (Figure 4c) and long branching is presumptively driven by sylvatic spillover seeding sporadic outbreaks followed by endemic circulation [77].

Genetic variation among emerging CHIKV strains may influence vector competence, viral fitness, and adaptation to varied settings, as was evident in the 2005–2006 Réunion Island outbreak with the acquisition of the E1:A226V mutation [80]. Initially, the Réunion strains had E1:226A and E1:226V later in the outbreak [3,91]. Independent convergent evolution led to the emergence of the E1:A226V amino acid substitution resulting in higher vector competence in *Ae. albopictus* [39]. Similarly, sequences from Madagascar, Seychelles, Mayotte, and Mauritius showed variations in the E1:226 amino acid, impacting local transmission and epidemic peaks [3]. The strain responsible for the Indian Ocean outbreak is believed to have originated from the Kenyan Mombasa outbreak in 2004. The most recent common ancestor (TMRCA) of the 2004 Kenyan genomes emerged in mid-2002, eventually diverging into the two variants responsible for the Lamu and the Mombasa outbreaks [35].

Strains collected in Central Africa between 2006 and 2019 belong to the ECSA Central African clade (Figure 4a) containing the E1:A226V mutation. Isolates from the 2020 outbreak in Chad clustered with the outbreak in DRC in 2019 but lacked the E1:226 mutation [92]. The presence of the E1:A226V mutation isolated from a French traveller returning from Madagascar in 2006 suggests the circulation of *Ae. albopictus*-associated-virus strains alongside *Ae. aegypti*-associated strains in Madagascar as early as 2006. Additionally, adaptive mutations for transmission by *Ae. albopictus*, such as the E1:A98T and the epistatic K211E mutations and several mutations in the E2 gene (D60G, R198Q, L210Q, I211T, K233E, K252Q), enhance viral fitness [39]. The E2:L210Q has been found to enhance CHIKV dissemination in *Ae. albopictus* present within sequences from the 2016–2019 outbreaks in Cameroon of the Central African clade [93]. Phylogenetic analysis suggests that the Central African clade containing the E1:A226V and E2:I211T mutations emerged around the same time as the IOL via convergent evolution [12,93]. At low prevalence in ECSA viruses, the E2:I211T is consistently present in viruses harbouring E1:A226V, with in vitro studies suggesting that it works epistatically to facilitate the effects of E1:A226V on infectivity [39].

Genomic analyses of CHIKV strains have also identified mutations such as E1:K211E and E2:V264A within the IOL, increasing infectivity and transmission in *Ae. aegypti* vectors. These mutations emerged between 2005 and 2008, likely originating in India [43]. Detected in the 2016 and 2018 Kenyan outbreaks, these mutations are believed to enhance the virus fitness for *Ae. aegypti* with no effect on *Ae. albopictus* vector competence [35].

To understand CHIKV transmission heterogeneity, we examined the spatial and temporal dispersal of CHIKV strains in and out of Africa using ancestral-state reconstruction (Figure 5). Our results show notable transmission patterns observed both within Africa and across continents. Within Africa, CHIKV spread from East Africa to Angola, South Africa, and Central Africa. In the late 1980s and mid-1990s, the virus exhibited transcontinental movement, moving from Central Africa to North America and from Angola to Brazil approximately two decades later. Concurrently, transmission from East Africa spread into the Indian subcontinent, notably highlighting the expansion of the ECSA genotype. Further analysis of the WA genotype showed no viral exchange between regions.

Our analysis of the IOL (Figure 5b) reveals initial exportation from East Africa to Comoros Island, followed by circulation to and among Réunion Island, Mauritius, Madagascar, and Mayotte Island. The international movement included transmission from East Africa into North America and the IOL to France, with subsequent spread to India between 2010 and 2015. A reintroduction event was observed from the Indian subcontinent into East Africa, which is also supported by the literature [43], followed by circulation into Northeast Africa after 2015.

The movement of infected individuals across borders has played a documented role in the global dissemination of CHIKV [2]. The spread of the ECSA genotype highlights the impact of travel, trade, and human migration on the CHIKV viral movement [94]. The first importation of the ECSA genotype into Brazil in September 2014 was closely related to the Angolan isolate from 1962 [94].

The IOL is thought to have evolved from the Mombasa strain of CHIKV [39], which is associated with morbidity and mortality, later spreading to the Indian Ocean basin and Asia, causing widespread epidemics [95,96]. The IOL outbreak led to several travel-associated cases and the first imported case of CHIKV into France from a traveller returning from the Comoros Islands [97]. The March 2005 outbreak on Réunion Island was traced back to a patient returning from Comoros Island, where the outbreak had begun in January 2005. Genetic analysis revealed that sequences from the early 2006 outbreak likely represented the ancestral genotype of the Réunion outbreak, sharing identical polymorphic sites with the ancestral ECSA lineage [3].

In the present study, genomic analysis shows an independent introduction of CHIKV from India to Kenya, with the 2016 outbreak genetically similar to Indian genomes. This is supported by the temporal introduction of the IOL from India to Kenya between 2010 and 2015, as seen in Figure 5b. The literature suggests a potential importation event across the Somali border [35]; however, without genomic data, we cannot correctly induce the likely route of introductions. We uncover evidence of reintroductions of the ECSA lineage (Figure 5a) into Africa from the Indian subcontinent. The 2018–2019 CHIKV epidemic that occurred in the eastern states of Sudan was caused by an independent introduction of CHIKV into the region from the Indian subcontinent [36]. The 2017–2018 Mombasa outbreaks indicate separate introductions [37]. These outbreaks spread to Eastern and Northern African countries, including Sudan (Figure 5b) and Djibouti in 2019. Our findings are consistent with the prior genomic investigations suggesting distinct introduction events with separate strains and temporal origins [43].

CHIKV transmission in West Africa typically occurs within an enzootic cycle involving primatophilic mosquito vectors and NHPs as a presumed natural reservoir [65,98], allowing viruses to persist in the natural environment [80]. However, the precise origin and dynamics of transmission of this cycle remain incompletely understood, necessitating further investigation [98]. The two circulating genotypes in Africa showed distinct separation [99], except for a single ECSA isolate from Senegal (Figure 4a). This separation highlights the complex interplay of ecological factors, socioeconomic situations, and viral transmission dynamics, supporting this study’s hypothesis that the heterogeneous nature of CHIKV transmission drives the ongoing circulation of the virus across Africa.

## 6. Vectors and Transmission

To reliably interpret CHIKV transmission dynamics, it is important to consider transmission routes, including the role of vectors. CHIKV primarily spreads through two main transmission cycles: the sylvatic and urban cycles [2,100]. In the sylvatic cycle, the virus is transmitted between nonhuman primates (NHPs) and forest-dwelling mosquitoes, occasionally spilling over into humans in close proximity to the forest [100,101,102,103]. This cycle is said to represent the ancestral state associated with periods of silence between outbreaks, attributed to the generational susceptibility of NHPs. Various mosquito species act as vectors in this cycle, including *Ae. aegypti formous*, *Ae. africanus*, *Ae. luteocephalus*, *Ae. neoafricanus*, *Ae. furcifer-taylori*, *Ae. dalzieli*, *Ae. vittatus,* and *Culex quinquefasciatus* [6,104,105,106]. Spillover of CHIKV from sylvatic cycles occurs via vector species that possess the ability to feed on both humans and NHPs, known as bridge vectors [100,103]. Primatophilic *Ae. africanus* and *Ae. furcifer* have been widely linked to the sylvatic transmission cycle in East–Central, Western, and Southern Africa [80,100]. During the 1976 epidemic in South Africa, *Ae. furcifer* was implicated as the primary vector of transmission with baboons as the likely NHP host [107]. *Aedes furcifer* was also implicated as a significant bridge vector for CHIKV transmission in West Africa [101,103]. In more rural settings, outbreaks are occasionally detected through the implementation of adequate routine surveillance and are related to the increase in sylvatic mosquito populations after periods of seasonal rainfall [64].

The adaptation of the virus to domesticated *Aedes* mosquito species, particularly *Ae. aegypti*, has provided an alternate vector and facilitated the introduction of the disease to previously unexposed human populations [108]. *Aedes aegypti*, also known as the yellow fever mosquito, is a container-breeding species primarily found in urban areas. This domesticated form is commonly found in tropical and subtropical regions in 167 countries globally [109]. The behavioural adaptation and ecology of *Ae. aegypti* to urbanization and deforestation make the species highly conducive for epidemic transmission [110,111].

The tropical ecoregions of the Afrotropical realm provide an ideal wet-dense ecosystem for both sylvatic and urban populations of both the peridomestic *Ae. aegypti* and, to a lesser extent, *Ae. albopictus* [12]. *Aedes albopictus* originating from Asia has spread across Central Africa; the first documented introduction into Africa occurred in 1989 via imported tires from Tokyo, Japan, to Cape Town, South Africa [66,112]. Its distribution has expanded to 126 countries worldwide and into a wide range of environments due to the species’ resilience in more temperate regions [109]. *Aedes albopictus* was the main vector responsible for CHIKV transmission in the 2007 Gabonese and 2011 CAR outbreaks [111]. *Aedes albopictus*, known for its invasive nature, exhibits a combination of zoophilic and anthropophilic behaviours that enable it to colonize new ecological niches and adapt to seasonal fluctuations. This adaptability is evident in its diverse breeding habitats, opportunistic feeding behaviours, and the ability to outcompete native species regardless of temperature conditions [24,113].

The global surge in CHIKV outbreaks and transmission has been strongly associated with the appearance of mutations in the virus permitting enhanced viral adaptation and fitness to existing and novel vectors, leading to increased transmission [92,114]. An important example of this adaptation is the E1:A226V mutation on the envelope 1 gene, facilitating the transmission of CHIKV by the highly invasive *Ae. albopictus* mosquitoes. The E1:A226V mutation has been detected in recent Central African CHIKV outbreaks, including the 2019 outbreak in the DRC [39,78]. The scenario of convergent evolution advocates for the colonization by *Ae. albopictus* in recent years, indicating significant implications for CHIKV transmission patterns [39]. This additionally suggests that CHIKV is adapting to the increased presence of *Ae. albopictus* in Africa, leading to new territories being colonized by this vector species and the potential for more frequent outbreaks in naive human populations [39,111].

Entomological surveillance yields valuable data on vector species prevalence, abundance, and behaviour, aiding in the understanding of arboviral transmission dynamics [114,115]. CHIKV transmission is closely associated with the passive dispersal of vectors via trade and migration, alongside increased environmental suitability, facilitating global vector spread [109,111,116]. In recent decades, these vectors have expanded to several areas previously devoid of *Aedes* species, with most new introductions attributed to vegetative eggs contained in timber and tire exportation [78]. Surveillance efforts have declined, with a 70% decrease in *Aedes* prevalence studies from 2009 to 2018 when compared to the previous decade [65]. Limited ecological and entomological studies exist for *Ae. aegypti* and *Ae. albopictus* in Guinea, Guinea-Bissau, Liberia, Sierra Leone, and Togo. Ongoing surveillance systems are crucial in detecting and controlling outbreaks, ensuring opportune public health responses [24,117]. Focus on routine surveillance systems and occurrence data should be prioritized, especially within countries with limited or no available data such as Guinea-Bissau, Togo, Chad, South Sudan, Ethiopia, Eritrea, and Somalia [114,118].

## 7. Challenges and Gaps in CHIKV Surveillance in Africa

Underreporting and underdiagnosis of CHIKV in Africa leads to an incomplete understanding of the true burden of the disease, especially in regions where multiple arboviruses cocirculate and exhibit similar clinical manifestations [3,36]. Socioeconomic factors and inadequate diagnostic infrastructure further exacerbate these challenges [36]. Inconsistent and nonstandardized case and outbreak reporting across different regions reduce the reliability, accuracy, and comparability of data, hindering the comprehensive assessment of CHIKV transmission dynamics.

This study identifies significant gaps in surveillance and diagnosis, emphasizing the need for enhanced genomic surveillance to better understand epidemiological patterns of CHIKV and interactions with other arboviruses [34,86,119]. Prospective studies should prioritize the identification of CHIKV isolation in potential host reservoir species, as an enhanced understanding of the enzootic, sylvatic cycle of the virus may prevent future outbreaks in regions where sylvatic transmission regularly occurs [120].

However, there is renewed hope of improved management and control of this infectious disease with the recent approval of a live-attenuated vaccine, lxchiq (VLA1553), by the Food and Drug Administration (FDA) in the United States, Canada, and Europe, as the vaccine has demonstrated a high immunogenicity of 98.9% after a single dose [121]. Despite this progress, public health systems must continue to enhance their efforts in tracking and controlling CHIKV outbreaks. The immunological state significantly impacts CHIKV transmission patterns. Periodic outbreaks may develop as a result of waning immunity or exposure to immunologically naive populations [13,69].

In Africa, neglected arboviruses like CHIKV can cause significant epidemics, yet our understanding of its impact, distribution, and viral heterogeneity remains limited. Historically, public health responses and resources have predominantly been directed towards malaria prevention and control [122]. Understanding the genetic diversity of CHIKV in Africa is essential for advising effective public health strategies for disease surveillance and outbreak control tailored to the continent and its regions. An assessment by the Resilience Against Future Threats through Vector Control (RAFT) research consortium in September 2022 revealed that most African countries have inadequate capacities for arbovirus surveillance and control [122]. Achieving global equity in genomic sequencing is vital for preparing for future pandemics. This goal requires targeted funding, distribution of sequencing technologies, increased training, networking, and informed public health policy decisions [86,119,123]. Effective leadership, collaboration, and adaption of existing surveillance systems are all essential for successful cross-border and continental disease control [86].

## 8. Conclusions

Urbanization and the rapid growth of human populations create an environment conducive to increased CHIKV transmission risk. Our findings underscore the complex dynamics of CHIKV transmission, highlighting its heterogeneous nature across the continent. Efforts to monitor and mitigate the movement of infected individuals are crucial in containing the spread of CHIKV and preventing further outbreaks. However, effectively addressing the CHIKV burden requires prioritized support for susceptible populations within epidemic regions. This includes improving access to healthcare, implementing effective vector control, and enhancing diagnostic measures to prevent the emergence and re-emergence of CHIKV.

Increased data collection is essential to better understand the epidemiological background and prevalence of CHIKV in Africa. Investing in research and surveillance systems enables the generation of comprehensive data, facilitating regional-based prevention and control strategies. Global collaboration and data sharing are essential in addressing the disease burden collectively. Through collaborative efforts, we can strengthen the public health response in at-risk communities, generate essential data, and promote research that effectively combats CHIKV in Africa. Coordinated action is indispensable to mitigate the impact of CHIKV and other emerging infectious diseases across the continent.

## Figures and Tables

**Figure 1 pathogens-13-00605-f001:**
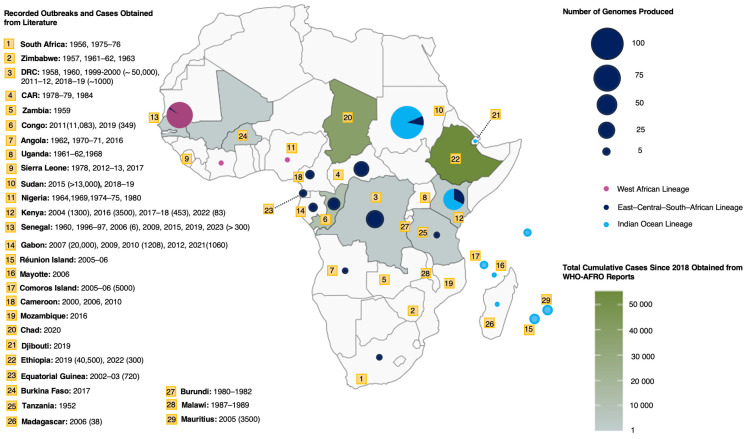
Distribution and surveillance of genome, cases, and outbreaks of CHIKV in Africa. The colour ramp of the map illustrates the geographic distribution of CHIKV cumulative and suspected cases from 2018 to 2024 obtained from the WHO-AFRO weekly reports [27]. Overlaid markers in yellow indicate the reported year of outbreaks and reported cases in parenthesis obtained from the literature [3,6,12,13,28,29,30,31,32,33,34,35,36,37,38,39,40,41,42,43,44,45,46,47,48,49,50,51,52,53,54,55,56,57,58,59,60], and the coloured circles show the lineages and the number of genomes generated in each country received from Bacterial and Viral Bioinformatics Resource Center (BV-BRC) [61], GISAID EpiArbo database [62], and NCBI’s GenBank databases.

**Figure 2 pathogens-13-00605-f002:**
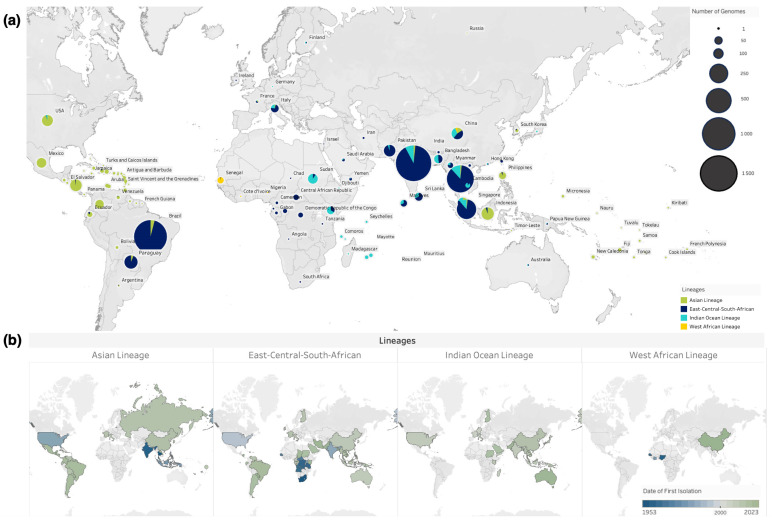
Global distribution of CHIKV lineages. (**a**) World map depicting the distribution and number of genomes generated for each CHIKV lineage. Circle sizes represent the number of CHIKV genomes produced per country and colours represent the proportion of different CHIKV lineages in each country. (**b**) Global map coloured by the date of first genome isolation for each CHIKV lineage, ranging from 1953 to 2023. Data source: Bacterial and Viral Bioinformatics Resource Center (BV-BRC) [61].

**Figure 3 pathogens-13-00605-f003:**
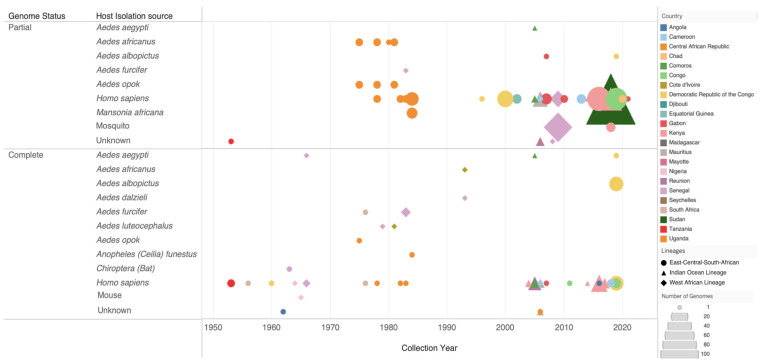
Overview of genomic sequencing efforts in Africa since the 1950s. Shapes are used to illustrate the different CHIKV genotypes: the circle represents the East–Central–South African genotype, the diamond represents the Indian Ocean lineage (IOL), and the square represents the WA genotype. The shapes’ sizes show the number of genomes produced for each isolation source in each African country, represented by different colours. The figure also depicts the type of genomes produced for each host species. Data source: Bacterial and Viral Bioinformatics Resource Center (BV-BRC) [61] and NCBI GenBank.

**Figure 4 pathogens-13-00605-f004:**
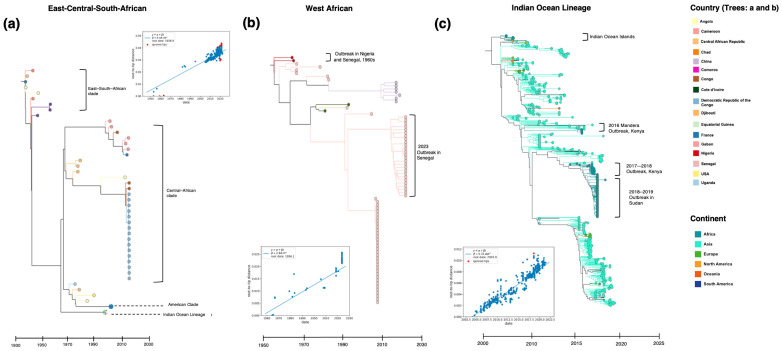
Maximum likelihood time-scaled tree illustrating the evolutionary relationships among the different circulating lineages in Africa, providing insights into the temporal aspects of CHIKV evolution and dispersal. (**a**) ECSA genotype: This genotype is depicted as showing two geographically distinct clades. (**b**) The WA genotype circulates within the West African region. (**c**) Indian Ocean lineage, showing recent outbreaks in Africa and the initial outbreaks on Indian Ocean islands, aggregated by continent.

**Figure 5 pathogens-13-00605-f005:**
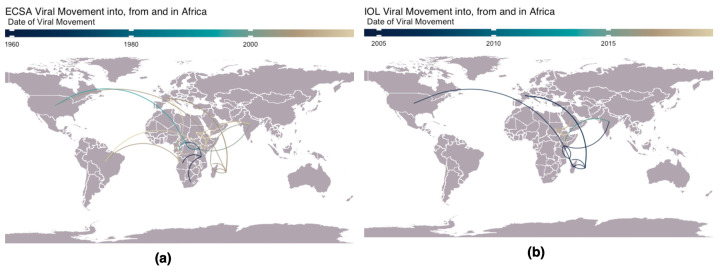
Transmission dynamics of CHIKV lineages worldwide and in Africa. World map illustrating the temporal and spatial CHIKV dissemination patterns in, out of, and within Africa. (**a**) The ECSA genotype and (**b**) IOL lineage.

## Data Availability

The data utilized in this study were sourced from publicly available repositories and publications. Genomic data were obtained from GenBank, Bacterial and Viral Bioinformatics Resource Center (BV-BRC) (https://www.bv-brc.org/, accessed on 12 January 2024) and GISAID EpiArbo database (https://www.gisaid.org/ accessed on 12 January 2024), under the GISAID identifiers EPI_ISL_19229161 to EPI_ISL_192291939. Outbreak and case data were retrieved from the relevant scientific literature and the World Health Organisation Regional Office in Africa (WHO-AFRO) (https://www.afro.who.int/health-topics/disease-outbreaks/outbreaks-and-other-emergencies-updates, accessed on 28 April 2024).

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
