# Peer review of "Understanding the Transmission Dynamics of the Chikungunya Virus in Africa"

_pathogens, 2024, doi:10.3390/pathogens13070605_

Round 1

Reviewer 1 Report

Comments and Suggestions for Authors

In this review entitled « Understanding the Transmission Dynamics of the Chikungunya Virus in Africa », Yajna Ramphal et al. provided a very nice picture of the current situation of Chikunugunya epidemic in Africa, taking in accout the world situation and the history of Chikungunya discovery and spread within the two emisphere South to North, East to West and West to East.

This review that included more than 120 articles is close to be exhaustive considering the continuous wave of publications that followed the spread of this virus including some reborn like the following paper from Bane et al. that probably explored the remains of arboviruses infection in a serosurvey performed following the Ebola epidemic in 2017-2018 in Southern Mali. (see below).

The struture of the article is clear and meaninfull. I have only minor comments

Comments:

Line 68-69 before the cost of chikungunya epidemic, I would like the author underline the existence of very severe cases of chikungunya disease, atypical in fact, within new-born of mothers that were viremic at the delivery. A fact first observed in La Réunion but with very few previous observation in the 70s in India and recently remembered in Nigeria (Sagy as et al.).  

The defence of the need to do a more exhaustive genotyping of this virus as it was done for SARS-CoV-2 is well presented and information obtained with the current data are of interest.

Finaly the chapter 6 about the challenge and gaps in CHIKV surveillance provide a very nice source of future research for the community.

The authors however should balance the sentence line 455 about the lack of effective vaccines for chikv as recently an attenuated CHIKV strain VLA1553 was used as a vaccine and a commercial one (Ixchik) si currently agreed by health authorities in USA, Canada and Europe for adults and his on the hedge of validation in Brazil for children and adolescent.

Recent reference of Chikungunya in Mali :

Seroprevalence of Arboviruses in a Malaria Hyperendemic Area in Southern Mali.

Bane S, Rosenke K, Feldmann F, Meade-White K, Diawara S, Keita M, Maiga O, Diakite M, Safronetz D, Doumbia S, Sogoba N, Feldmann H. Am J Trop Med Hyg. 2024 Jun 4:tpmd230803. doi: 10.4269/ajtmh.23-0803. Online ahead of print. PMID: 38834052

Chikungunya virus antepartum transmission and abnormal infant outcomes in a cohort of pregnant women in Nigeria.

Sagay AS, Hsieh SC, Dai YC, Chang CA, Ogwuche J, Ige OO, Kahansim ML, Chaplin B, Imade G, Elujoba M, Paul M, Hamel DJ, Furuya H, Khouri R, Boaventura VS, de Moraes L, Kanki PJ, Wang WK. Int J Infect Dis. 2024 Feb;139:92-100. doi: 10.1016/j.ijid.2023.11.036. Epub 2023 Dec 4.

Author Response

Thank you very much for taking the time to review our manuscript. We sincerely appreciate your valuable feedback and constructive comments. Your insights have greatly contributed to enhancing the quality of our work. Please find our detailed responses to each of your comments below. We have addressed all the points raised and incorporated the corresponding revisions into the manuscript. These changes are highlighted in yellow in the re-submitted files.

Comment 1: This review that includes more than 120 articles is close to be exhaustive considering the continuous wave of publications that followed the spread of this virus including some reborn like the following paper from Bane et al. that probably explored the remains of arboviruses infection in a serosurvey performed following the Ebola epidemic in 2017-2018 in Southern Mali. 

Response 1: We appreciate the acknowledgement of the comprehensiveness of our review and your suggestion to include the recent study by Bane et al. on arbovirus infections in Southern Mali. We have reviewed the paper by Bane et al. (2024) and find that this study adds valuable insights into existing and undetected CHIKV circulation in post-epidemic settings. Therefore, we have incorporated its findings into our manuscript on page 4, paragraph 2, lines 149-150 to help better understand the burden of CHIKV on the African continent.

Updated text "A recent study on arboviruses in Southern Mali revealed a CHIKV seropositivity rate of 31.2% post-rainy season in 2015".

Comment 2: Line 68-69 before the cost of chikungunya epidemic, I would like the author underline the existence of very severe cases of chikungunya disease, atypical in fact, within new-born of mothers that were viremic at the delivery. A fact first observed in La Réunion but with very few previous observation in the 70s in India and recently remembered in Nigeria (Sagy as et al.).

Response 2: Thank you for highlighting the importance of addressing severe cases of Chikungunya in newborns. We have added information about the antepartum and intrapartum infection transmission of CHIKV and its association with adverse birth outcomes, as observed in a recent cohort study in Nigeria from the references provided. This addition can be found on page 2, paragraph 2, lines 67-71.

Updated text: “High rates of mother-to-child transmission have been observed during outbreaks in India, the Americas, and Reunion Island, leading to neonatal disease with significant impacts on infant health. Recent evidence from a cohort study in Nigeria (2019-2022) has highlighted significant associations between CHIKV infections during pregnancy and adverse birth outcomes [14,15].”

Comment 3: The defence of the need to do a more exhaustive genotyping of this virus as it was done for SARS-CoV-2 is well presented and information obtained with the current data are of interest.

Response 3: We are grateful for your positive feedback on our discussion regarding the necessity for more comprehensive genotyping of CHIKV. We hope that our discussion encourages further research and collaboration in this critical area, contributing to enhanced surveillance and public health response strategies for CHIKV.

Comment 4: Finaly the chapter 6 about the challenge and gaps in CHIKV surveillance provide a very nice source of future research for the community.

Response 4: Thank you for your kind words about Chapter 6. We are glad you found the discussion on CHIKV surveillance challenges and gaps valuable for future research directions.

Comment 5: The authors however should balance the sentence line 455 about the lack of effective vaccines for chikv as recently an attenuated CHIKV strain VLA1553 was used as a vaccine and a commercial one (Ixchik) si currently agreed by health authorities in USA, Canada and Europe for adults and his on the hedge of validation in Brazil for children and adolescent.

Response 5: Acknowledging this important and recent advances in CHIKV vaccines, we have revised the manuscript on page 11, paragraph 3, line 468-471 to include the use of the VLA1553 attenuated CHIKV strain and the commercial availability of Ixchiq. The updated text now reflects their regulatory status in the USA, Canada, and Europe.

Updated text: “However, there is renewed hope of improved management and control of this infectious disease with the recent approval of a live-attenuated vaccine, lxchiq (VLA1553), by the Food and Drug Administration (FDA) in the United States, Canada, and Europe, as the vaccine has demonstrated a high immunogenicity of 98.9% after a single dose [124].”

Reviewer 2 Report

Comments and Suggestions for Authors

The manuscript by Ramphal and colleagues provide a valuable review of the epidemiology and genomics of chikungunya virus infection throughout Africa, with emphasis on the need for enhanced surveillance to characterize shifts in genotype in association with things like enhanced vector competence.  Clearly this is an important pathogen and this review is timely.  The manuscript is generally well written and presented.  There are several sections of redundance (e.g., definition of the different viral clades), but that is a minor criticism.  Indeed, I do not have substantive criticisms or suggestions except to point out that the manuscript would benefit from another round of editing for language.  As examples:

Line 37: eighteenth and nineteenth [centuries I assume?]

Line 47: suggest “some of the earliest human cases” (since you are referring to outbreaks)

Line 97: what does “long-term genomic sequencing capacity” mean – sustained, continuous?

Line 258: “As a result, providing important information ..” – incomplete sentence

Line 444: “particularly in regions of co-circulation of [other] arboviruses.”

One final comment: the last paragraph is perhaps too much of a cheerleading/advertising statement for CLIMADE and should probably be subdued.

Comments on the Quality of English Language

Several examples of problem with sentence structure described in review.

Author Response

Thank you very much for taking the time to review our manuscript. We sincerely appreciate your thoughtful and constructive feedback. Your insights have been invaluable in helping us refine and improve our work. Below, you will find our detailed responses to each of your comments. We have carefully addressed all the points raised and have made corresponding revisions to the manuscript. These changes are highlighted in yellow in the re-submitted files.

Comment 1: The manuscript by Ramphal and colleagues provide a valuable review of the epidemiology and genomics of chikungunya virus infection throughout Africa, with emphasis on the need for enhanced surveillance to characterize shifts in genotype in association with things like enhanced vector competence.  Clearly this is an important pathogen and this review is timely.  The manuscript is generally well written and presented. 

Response 1: We are pleased that you find our review on CHIKV epidemiology and genomics valuable and timely. Your recognition of the manuscript's clarity and relevance is greatly appreciated.

Comment 2: There are several sections of redundancy (e.g., definition of the different viral clades), but that is a minor criticism. I do not have substantive criticisms or suggestions except to point out that the manuscript would benefit from another round of editing for language.

Response 2:  Thank you for pointing out the redundant sections and areas that require clarification. We have streamlined the sections to enhance clarity and avoid repetition by removing repetitive mentioning of genotypes (removed from page 4, paragraph 5, line 173-174) as this was introduced in the background chapter of the review. We have conducted an additional round of revisions to ensure the manuscript meets the highest standards of readability. We list below the edits you highlighted: 

Line 37: Clarify the timeframe: "eighteenth and nineteenth centuries."

Updated text “Clinical studies from the eighteenth and nineteenth centuries, along with molecular clock analyses of contemporary CHIKV genomes, suggest that this virus existed for 300 to 500 years before the first isolation in 1953”.

Line 47: Revise to: "suggest 'some of the earliest human cases' (since you are referring to outbreaks)."

Updated text “The Chikungunya fever outbreak in the Newala district of Tanzania in 1952 and 1953 holds notable significance as it suggests some of the earliest documented cases in humans and marks the first isolation of the virus”.

Line 97: Specify: "What does 'long-term genomic sequencing capacity' mean – sustained, continuous?"

Updated text lines 99-102 “This initiative, with global collaborators, aims to form a pan-African coalition to create localised knowledge and technologies to fill crucial knowledge gaps in disease trans-mission, improve the genomic representation of climate-amplified diseases, and develop sustained genomic sequencing capacities”.

Line 258: Complete the sentence: "As a result, providing important information..."

Updated text lines 266-268 “This effort provides essential information for implementing effective control and preventive measures against CHIKV and other vector-borne diseases in Africa”.

Line 444: Clarify: "particularly in regions of co-circulation of other arboviruses."

Updated text line 455-457 “Under-reporting and under-diagnosis of CHIKV in Africa leads to an incomplete understanding of the true burden of the disease, especially in regions where multiple arboviruses co-circulate and exhibit similar clinical manifestations”.

Comment 3: One final comment: the last paragraph is perhaps too much of a cheerleading/advertising statement for CLIMADE and should probably be subdued.

Response 3: We have toned down this paragraph to reflect the need for a united and collaborative effort to mitigate CHIKV outbreaks without highlighting CLIMADE.

Updated text line 503-507 “Global collaboration and data sharing are essential in addressing the disease burden collectively. Through collaborative efforts, we can strengthen the public health response in at-risk communities, generate essential data, and promote research that effectively combats CHIKV in Africa. Coordinated action is indispensable to mitigate the impact of CHIKV and other emerging infectious diseases across the continent”.

Comment 4: Several examples of problem with sentence structure described in review.

Response 4: We appreciate your suggestion to refine the manuscript’s language and have undertaken an additional round of editing to address sentence structure and other language issues to ensure clarity and readability.